# Nutritional Interventions for Patients with Melanoma: From Prevention to Therapy—An Update

**DOI:** 10.3390/nu13114018

**Published:** 2021-11-11

**Authors:** Marianna Pellegrini, Chiara D’Eusebio, Valentina Ponzo, Luca Tonella, Concetta Finocchiaro, Maria Teresa Fierro, Pietro Quaglino, Simona Bo

**Affiliations:** 1Department of Medical Sciences, Division of Endocrinology, Diabetes and Metabolism, University of Torino, 10126 Torino, Italy; marianna.pellegrini@unito.it (M.P.); chiara.deusi@gmail.com (C.D.); valentina.ponzo@unito.it (V.P.); simona.bo@unito.it (S.B.); 2Department of Medical Sciences, Dermatologic Clinic, University of Torino, 10126 Torino, Italy; luca.tonella.trials@gmail.com (L.T.); mariateresa.fierro@unito.it (M.T.F.); 3Dietetic and Clinical Nutrition Unit, “Città della Salute e della Scienza” Hospital, 10126 Torino, Italy; ettafinocchiaro@gmail.com

**Keywords:** melanoma, diet, nutrition, obesity

## Abstract

Melanoma is an aggressive skin cancer, whose incidence rates have increased over the past few decades. Risk factors for melanoma are both intrinsic (genetic and familiar predisposition) and extrinsic (environment, including sun exposure, and lifestyle). The recent advent of targeted and immune-based therapies has revolutionized the treatment of melanoma, and research is focusing on strategies to optimize them. Obesity is an established risk factor for several cancer types, but its possible role in the etiology of melanoma is controversial. Body mass index, body surface area, and height have been related to the risk for cutaneous melanoma, although an ‘obesity paradox’ has been described too. Increasing evidence suggests the role of nutritional factors in the prevention and management of melanoma. Several studies have demonstrated the impact of dietary attitudes, specific foods, and nutrients both on the risk for melanoma and on the progression of the disease, via the effects on the oncological treatments. The aim of this narrative review was to summarize the main literature results regarding the preventive and therapeutic role of nutritional schemes, specific foods, and nutrients on melanoma incidence and progression.

## 1. Introduction

The world-wide distribution of melanoma is characterized by higher incidences in Australia, New Zealand, and North America [1], together with an increase in the incidence rates in most European countries [2,3]. 

Risk factors for melanoma are divided into two groups [4,5,6,7]. The first group is represented by intrinsic or constitutional factors: familiar and genetic factors (familiarity for melanoma cases, mutations of susceptibility genes, such as Cyclin-dependent kinase 4 (CDK4)) and skin features (number of nevi, presence of atypical nevi, and phenotype/phototype). The phenotype identifies the color of skin, eyes, and hair. The phototype identifies skin reactivity to UV radiation, risk of burns and ability to tan; it is graded according to the Fitzpatrick scale from type-1 (always burns, never tans) to 6 (never burns) [8]. The characteristics associated with a greater risk of developing melanoma are represented by the presence of light skin, red or blond hair, poor ability to tan, and susceptibility to sunburn [4,5,6,7,8].

The second group is represented by extrinsic or acquired factors, among which the most important is represented by exposure to natural and/or artificial sources of ultraviolet (UV) rays. The onset of melanoma is related to intense but intermittent sun exposure, and the development of sunburn, particularly in childhood and/or adolescence. Only lentigo maligna-melanoma depends on the cumulative dose of UV rays.

The increase in melanoma is mainly driven by thin lesions, whilst thick melanomas with poor prognosis increased more slowly due to the improvement in early diagnosis [9]. The introduction of targeted treatments, such as anti-BRAF therapies and check point inhibitors, have largely improved our potential to manage advanced metastatic melanoma patients, inducing a significant improvement in response rates and overall survival (OS). The same drugs have also shown relevant activity in adjuvant settings [10,11].

Increasing evidence has suggested the importance of nutritional factors in the management of melanoma. This review summarizes the main literature data related to the role of nutritional factors on melanoma prevention or modulation.

## 2. Materials and Methods

A targeted literature search was conducted in PubMed, the Cochrane Library, EMBASE, and CINAHL databases to identify studies on nutritional interventions in relation to melanoma. A combination of terms related to melanoma and diet/nutrition/food/nutrients were used. English-language studies were considered; no restrictions were imposed. Manual searching the references of the studies and reviews on the field was performed to augment the search strategy. The research articles were considered with the following scale of priority: systematic reviews and meta-analyses, randomized controlled trials (RCTs), human observational studies, case series. In the case of unavailability of human studies, animal studies and in vitro studies were also considered.

## 3. Obesity and Melanoma: A Controversial Question

Obesity was previously associated with both increased cancer risk and worse prognosis and outcomes in various neoplasms [12]; however, data on melanoma are not conclusive [13,14]. Higher body mass index (BMI), body surface area, height [15,16,17], and obesity-related insulin resistance [15] have been reported to increase the risk for cutaneous melanoma; conversely, bariatric surgery seems to be protective [18]. Two meta-analyses of prospective observational studies [12] or cohort and case–control studies [19], evaluating 8278 and 12,355 patients with melanoma, respectively, reported a positive association between increased BMI and malignant melanoma risk in males (risk ratio (RR) = 1.17, 95%CI 1.05–1.3011; 1.31, 95%CI 1.22–1.4118). By contrast, a null association between BMI and melanoma risk was found in females [19], but a possible confounder may be a self-limited sun exposure among overweight or obese females resulting in less intense sunbathing. Worse outcomes after resection, and increased Breslow thickness, were reported in patients with both melanoma and obesity, but confounding factors including gender, menopausal status and ultraviolet exposure should be considered [13]. Obesity could promote the development and progression of melanoma as well as impairing the chemotherapeutic response via several mechanisms. Excessive adipose tissue induced elevated levels of hormones and growth factors (such as insulin, insulin-like growth factor (IGF)-1, and sex steroid hormones), adipokine imbalances, and a chronic state of low-grade inflammation [20]. In animal and in vitro studies, the obesity-related high levels of adipokines, such as leptin and resistin, impaired the therapeutic efficacy of dacarbazine (DTIC) towards melanoma regression and induced a drug-resistant phenotype by upregulating fatty acid synthase (FASN), caveolin (Cav)-1, and P-glycoprotein (P-gp) [21,22]. Conversely, caloric restriction and the treatment with orlistat, an anti-obesity drug, determined—both in animal models and in vitro—a reduction in melanoma progression, likely related to the normalization of resistin and leptin serum levels secondary to adiposity reduction, with a lower synthesis of FASN, Cav-1, and activated protein kinase B (Akt) levels [23]. 

However, as for other cancers, an “obesity paradox”, that is, an outcome improvement in patients with a higher BMI, was reported for melanoma [13]. A few studies compared clinical outcomes, such as the progression-free survival (PFS), the OS, the objective response rate (ORR), and the time to treatment failure (TTF) in melanoma patients receiving immunotherapy, targeted therapy, or chemotherapy according to their BMI. With respect to normal weight, patients with overweight or obesity treated with immunotherapy showed an improved PFS [24,25,26,27], OS [24,25,26,27,28], ORR, and TTF [26], and these associations were predominantly driven by male gender [24,25,26]. Improved PFS and OS were found in male patients with obesity receiving targeted therapy, but not chemotherapy [24]. However, other study groups did not find significant associations between BMI and PFS [29,30,31], OS [29,30,31], disease control rate (DCR) [30], or toxicities [29,31] in patients with metastatic melanoma receiving immunotherapy. 

## 4. The Role of Nutrition in the Prevention and Therapy of Melanoma

The importance of nutrition in the prevention and management of a wide range of diseases, including cancer, is well known, since the anti-inflammatory, immunomodulatory, and antioxidant properties of foods have been demonstrated [32,33,34]. Healthy eating habits, characterized by an elevated consumption of natural un-processed foods, fiber, potassium, and antioxidants, an increased ratio of polyunsaturated (PUFA)/monounsaturated (MUFA) to saturated fatty acids (SFA), and a low intake of animal proteins, refined grains, sodium, and empty calories (such as added sugars and alcohol) seem to be protective against the risk of melanoma [35]. Hence, specific dietary patterns, foods, spices, or beneficial ‘natural’ supplements derived from food (i.e., nutraceuticals) may be potential additional tools during oncological treatments [36,37]. The available evidence on the association between melanoma and nutrition are described below and summarized in Table 1, considering both its preventive and potential “therapeutic” role, ranging from whole nutritional schemes, to individual foods and their components, up to supplements.

### 4.1. Diets

#### 4.1.1. Mediterranean Diet

The Mediterranean Diet (MeD) is a dietary pattern characterized by a high consumption of plant foods, such as vegetables, legumes, fruit, cereals, and olive oil, and moderate-to-low consumption of fish, meat, dairy products, added sugars, and wine assumed during meals [38]. This nutritional scheme contains high amounts of MUFA, n3-PUFA, polyphenols, vitamins, minerals, and flavonoids, which exert anti-inflammatory, antioxidant, and immunomodulatory effects [39]. High adherence to the MeD was found to be correlated with lower incidence of melanoma (HR = 0.72, 95%CI 0.54–0.96, *p* = 0.02 for high compared with low adherence scores to the MeD) in a cohort of 67,332 French women, who developed 404 cases of melanoma during a 15-year follow-up, without any difference within sex or age groups [40].

Among single dietary components, an inverse borderline association between vegetable intake and risk of melanoma was found (HR = 0.87, 0.71–1.06), but not for other typical MeD foods [40]. However, it must be highlighted that the protection against skin cancer was found to be related to a high level of adherence to the MD as a whole, and not to the consumption of a single food [40].

Conversely, Malagoli et al. [35] failed in finding any correlation between MeD scores and melanoma, except for women aged < 50 years, in whom higher MeD adherence was associated with lower melanoma risk, suggesting a potential role for estrogen hormones [41,42].

#### 4.1.2. DASH Diet

The DASH (Dietary Approaches to Stop Hypertension) diet is characterized by a high amount of whole grains, fruits, vegetables, and low-fat dairy products; it was designed for the treatment of arterial hypertension [43] but it showed protective actions against the development of colorectal cancer and the related mortality [44]. An Italian population-based case–control study found an inverse association between high DASH adherence and melanoma risk in both sexes (p trend = 0.03), but in stratified analyses, the association was evident in women only (OR = 0.58, 95%CI 0.30–1.13; *p* = 0.04), with a stronger inverse association in younger (<50 years) women (OR = 0.80, 95%CI 0.70, 0.93; *p* = 0.04) [35]. 

### 4.2. Foods and Nutrients

Several studies have investigated the causal link between specific foods and melanoma risk. A systematic review of case–control and cohort studies showed a trend of reduced risk for melanoma with high intake of vegetables (by 40–57%) and fruits (by 34–46%) [45]. However, the highest-quality and largest cohort studies analyzed in the review, including 162,078 US adult women [46] and 50,575 Norwegian men and women aged 16–56 years [47], failed to find any correlation between fruit and vegetable consumption and melanoma risk, while a protective role was found in smaller, low-quality case–control studies only.

An Italian population-based case–control study involving 380 cases of melanoma and 719 age- and sex-matched controls [48] observed a positive direct association between melanoma risk and increased consumption of cereals and cereal products (OR = 1.32, 95%CI 0.89–1.96), sweets (OR = 1.22, 0.84–1.76), particularly chocolate and candy bars (OR = 1.51, 1.09–2.09), and cabbages (OR = 1.51, 1.09–2.09). Conversely, an inverse association with disease risk was found for the intake of legumes (OR = 0.77, 0.52–1.13), olive oil (OR = 0.77, 0.51–1.16), eggs (OR = 0.58, 0.41–0.82), and onion and garlic (OR = 0.80, 0.52–1.14) [48]. No relationship was observed with beverage consumption. Interestingly, there was a linear association in all categories except for olive oil, whose association disappeared with very high intake (>60 g/day). Furthermore, in subjects with higher adherence to the MeD, none of the food items seemed to have a clear direct association with melanoma risk, except for ‘processed meat’ and some types of ‘sweets’ and ‘white wine’, while in subjects with lower MeD adherence, the intake of meat and meat products (especially red meat), cheese, and mushrooms appeared to be a risk factor for melanoma. Those findings suggested that a high adherence to the MeD might counteract the unfavorable effects of certain foods [48].

#### 4.2.1. Red and Processed Meat

Few studies investigated the association between red and processed meat and the risk for melanoma. Surprisingly, in two US white prospective cohorts of 75,263 women from the Nurses’ Health Study (NHS, 1984–2010) and 48,523 men from the Health Professionals Follow-up Study (HPFS, 1986–2010), including 1318 melanoma cases, red and processed meat showed a protective effect on melanoma development (HR = 0.81, 95%CI 0.70–0.95, *p* = 0.002 for the highest vs. the lowest quintile of intake), independently of other known melanoma risk factors and potential confounders [49]. This beneficial effect may be related to the compounds contained in red and processed meat, such as retinol, with an inhibitory effect on promoting melanoma by restoring the “normal” functions (differentiation) of melanocytes [49,50], and nicotinamide (NAM), a niacin derivative, with a chemo-preventive effect via an immune response at pharmacological doses in vivo [51]. These finding were consistent with a previous large prospective study (*n* = 500,000), where processed meat (highest vs. lowest quintile of intake, HR = 0.82, 0.71–0.96) but not red meat were inversely associated with melanoma [52]. Conversely, a null association between red and processed meat intake and melanoma was found in two small-size case–control studies including 278 US [53] and 59 Italian [54] patients with melanoma. Meat and processed meat can be related to melanoma development through their contents of Advanced Glycation End Products (AGEs), the amount of which was shown to increase after broiling or frying [55]. AGEs have been shown to increase melanoma cell proliferation and migration, tumor growth, and metastasis through the interaction with their receptor (RAGE) on the melanoma cell surface. The blockade of the AGE/RAGE axis prevents tumor growth and inhibits angiogenesis [56].

#### 4.2.2. Alcohol

A meta-analysis of 14 case–control and 2 cohort studies on 6251 cases of melanoma [57] found a 20% increased risk for alcohol drinking compared with none/occasional drinking (RR = 1.20, 95%CI 1.06–1.37, *p* = 0.006). The adjustment for sun exposure (reported in 10 studies) led to a loss of statistical significance (RR = 1.15, 0.94–1.41). Furthermore, the relationship was linear with increasing alcohol intake in drinkers, with an estimated 55% excess risk for >50 g of daily ethanol intake [57]. A prospective cohort study on 59,575 white post-menopausal women followed-up for 10.2 years [58] reported a 64% increased melanoma risk in those who consumed seven or more drinks/week compared to non-drinkers (HR = 1.64, 1.09–2.49, *p* = 0.001). Moreover, the type of alcoholic beverage was significantly associated with melanoma risk, with particularly strong associations found for the consumption of white wine (HR = 1.52, 1.02–2.27) and liquor (HR = 1.65, 1.07–2.55) [58]. Another study analyzed three large US cohorts containing a total of 210,252 adults and identified 1374 cases of melanoma during a mean follow-up of 18.3 years, reporting a significant association between higher alcohol intake and incidence of invasive melanoma (HR = 1.14, 1.00–1.29, *p* = 0.04 per each drink/day), which was stronger in relatively UV-spared sites, such as trunk lesions (HR = 1.73, 1.25–2.38, *p* = 0.02) [59]. The authors confirmed the association between white wine consumption and the increased melanoma risk (HR = 1.13, 1.04–1.24, *p* < 0.01 per drink/day) [59]. Although unclear, this correlation could be promoted via acetaldehyde, by causing DNA damage and photosensitizing skin to UV radiation. Indeed, in white wine, acetaldehyde is both pre-existing (at similar levels of red wine, whose antioxidants may offset these risks) and produced by the conversion of ethanol after ingestion [59]. A pooled analysis of eight case–control studies in women (1886 melanoma cases and 2113 controls) confirmed a positive association between alcohol consumption and risk of melanoma after adjustment for sun exposure (pooled OR = 1.3, 1.1–1.5) [60]. Interestingly, the risk was increased for melanoma of the trunk (pooled OR = 1.5, 1.1–1.9) [60]. The more relevant carcinogenic effect of alcohol in relatively UV-spared sites supported the hypothesis that melanoma might have a different etiology depending on its anatomical site [59].

#### 4.2.3. Coffee and Tea

In 2016, a negative association between coffee consumption and melanoma risk was reported by both a meta-analysis of observational studies (927,173 adults, 3787 melanoma cases) [61] and a meta-analysis of case–control (846 melanoma patients and 843 controls) and cohort studies (844,246 adults and 5737 melanoma cases) [62]. The former reported a RR of 0.75 (95%CI 0.63–0.89, *p* = 0.001) for melanoma among regular coffee drinkers (>1 to ≥4 cups/day depending on the study) [61]; the latter showed a pooled RR of 0.81 (0.68–0.97, *p* = 0.003) for the highest versus the lowest coffee intake (≥4–7 to <1 cups/day depending on the study) [62]. Three recent cohort studies confirmed these results [63,64,65]. Lukic reported a reduced melanoma risk with both low-moderate (>1–3 cups/day, HR = 0.80, 0.66–0.98) and high-moderate consumption of filtered coffee (HR = 0.77, 0.61–0.97, for >3–5 cups/day compared to ≤1 cup/day) in Norwegian women [63]. Park reported a 38% reduced risk among non-white adults (HR = 0.72, 0.52–0.99, *p* = 0.002 for ≥4 cups/day vs. none) [65]. Caini found an inverse association between coffee consumption and melanoma risk among men only, from the European Prospective Investigation into Cancer and Nutrition (EPIC) cohort (HR = 0.31, 0.14–0.69, *p* = 0.001 for the highest quartile of consumption vs. non-consumers) [64]. 

The studies evaluating the association between tea consumption and melanoma risk showed contrasting results. No significant associations were reported both in a prospective cohort study of 35,369 postmenopausal US women [66] and in the EPIC cohort involving over 500,000 adults [64], while a smaller case–control study involving 609 adults found a protective effect for daily tea drinking (unspecified quantity) (OR = 0.42, 0.18–0.95; *p* = 0.025) [67].

#### 4.2.4. Polyunsaturated Fatty Acids

PUFAs are contained in plant- and fatty fish-derived oils and include omega-6 (n-6) and omega-3 (n-3) fatty acids, displaying a pro-inflammatory and anti-inflammatory role, respectively [68]. Both n-3 PUFAs, particularly eicosapentaenoic acid (EPA) and docosahexaenoic acid (DHA), and an advantageous n-3/n-6 ratio showed anti-cancer properties [69,70] by regulating tumor initiation, progression, and metastasis that, together, impact OS [69,70], especially in colon-rectal cancer [69,71].

Few epidemiological and case–control studies correlated higher dietary intake of PUFAs [72], and especially n-3 PUFAs [67,73], with a reduced risk of melanoma. Conversely, higher n-6 PUFAs intake was associated with an increased risk of melanoma in both the NHS and HPFS cohorts (HR = 1.20, 95%CI 1.02–1.41; *p* = 0.03 for highest vs. lowest quintiles) [74]. Similarly, a relationship with an increased prevalence of thick melanomas was found in an Australian prospective study on 634 patients (Prevalence Ratio (PR) = 1.40; 1.01–1.94, for highest vs. lowest tertiles of intake of the “meat-fish-fat” dietary pattern, which was characterized by high consumption of meat, fish, seafood, processed meat, eggs, peas, beans, and solid fats) [75]. Furthermore, in a large Norwegian cohort (*n* = 50,757 individuals), the intake of cod liver and PUFAs both led to a three-fold and four-fold increased risk of melanoma, respectively, but in women only [47]. However, the largest melanoma genome-wide association study, with Mendelian randomization on 12,874 cases and 23,023 controls, concluded that the effect of increased PUFAs levels on melanoma risk was either zero or very small [76]. Other studies failed in reporting any role of PUFA in melanoma development [53,54,77,78].

#### 4.2.5. Citrus Fruits and Vitamin C

A particular focus has been placed on melanoma risk and the consumption of citrus fruits, i.e., orange, grapefruit, tangerines, tangelos, and their respective juices, which are sources of vitamin C and psoralens. In vitro, vitamin C showed toxic, pro-apoptotic and anti-proliferative properties against melanoma cells; on the contrary, psoralens, a group of naturally occurring furocoumarins, are known to sensitize the skin to UVA radiations with photocarcinogenic effects [79]. Surprisingly, in the two large NHS and NHS II women cohorts, a higher melanoma risk (RR = 1.43, 95%CI 1.01–2.00, *p* = 0.05) was reported with greater intakes (175 mg/day) of vitamin C from foods compared with lower intakes (<90 mg/day), but not from supplements [46]. A significant positive dose–response relationship with frequency of orange juice consumption was reported (*p* = 0.008) [46]. The high content of other photoactive compounds in vitamin C–rich foods, but not in supplements, might be the reason for these contrasting results [46]. Among 105,432 individuals from the NHS and HPFS cohorts, 1840 incident melanomas were reported over 24–26 years of follow-up [79]. The risk of melanoma increased from <2 citrus/week consumption to ≥1.6 times/day consumption (HR = 1.36, 1.14–1.63, *p* < 0.001). Grapefruit consumption showed the highest risk (HR = 1.41, 1.10–1.82, *p* < 0.001 for 3 times/week vs. never) [79]. On the other hand, in a population-based case–control study in Northern Italy, involving 380 melanoma patients and 719 matched controls, vitamin C intake showed a protective effect (OR = 0.86, 0.65–1.15 and OR = 0.59, 0.37–0.94, for the intermediate and highest categories of vitamin C dietary intake), with a stronger association in younger females and in subjects with skin phototypes II and III [80]. Finally, a study involving 388,467 US adults with 3894 incident melanomas during 15.5-year follow-up failed in finding any overall relationship between citrus intake and melanoma risk [81]. However significant trends toward increased melanoma risk and whole citrus fruit consumption were found among subjects with higher estimated exposure to UV radiation, and of >60 years of age [81]. Therefore, at present, this association remains highly controversial and even the relevant number of studies does not allow the obtaining of conclusive results.

#### 4.2.6. Vitamin D

Vitamin D is a fat-soluble prohormone whose active form, 1,25-hydroxyvitamin D (1,25(OH)_2_D), is produced after hydroxylation in the liver and kidney [82]. Only about 10–20% of required vitamin D is derived from the diet, since about 80–90% is obtained from photosynthesis in the skin via the action of UV-B radiation [83]. The skin is not only the site of vitamin D synthesis, but also a target tissue for biologically active metabolites of vitamin D [84]. The 1,25(OH)_2_D binds to the nuclear vitamin D receptors (VDR), which are present in most tissues and cells in the body, including keratinocytes and melanocytes [85]. Besides maintaining serum calcium and phosphorus homeostasis, it plays a role in modulating the immune, cardiovascular, and inflammatory systems, as well as in participating in the regulation of cell proliferation, differentiation, and apoptosis [82]. The biological anti-cancer effects of 1,25(OH)_2_D include the induction of cell-cycle arrest, the stimulation of apoptosis, and the inhibition of angiogenesis and metastasis [86]. Indeed, the interpretation of vitamin D’s effects on the risk of skin cancer is complex because the exposure to UV radiation is both the most important source of vitamin D production, with its potential anticancer properties, and an environmental risk factor implicated in skin carcinogenesis [87] through the formation of cyclobutane pyrimidine dimers bringing DNA replication-dependent mutagenicity [88] and via the production of reactive oxygen species (ROS), leading to premature skin aging and lipid peroxidation [89].

A recent systematic review and meta-analysis of prospective studies found a null association between 100 IU/day vitamin D intake and melanoma development either for vitamin D intake from diet (RR = 1.01, 95%CI 0.99–1.03, *p* = 0.71), or from supplements (RR = 1.00; 0.96–1.03, *p* = 0.93) [87]. Nonetheless, a population-based randomized trial enrolling 154,897 US adults reported an increased, though not significant, risk of melanoma among men with a vitamin D intake in the highest quartile (>13.5 µg/day) (HR = 1.26, 0.93–1.70, *p* = 0.14), and a reduced risk in women (HR = 0.63, 0.41–0.96, *p* = 0.03) [90]. Similarly, a trial involving 36,282 postmenopausal women, randomized to receive 1000 mg of elemental calcium plus 400 IU of vitamin D daily (intervention group) or placebo for a mean 7-year follow-up period, reported a reduced risk of melanoma in the intervention group (HR = 0.43, 0.21–0.90; *p* = 0.038) [91]. Conversely, an Italian population-based case–control study, involving 380 melanoma cases and 719 healthy controls, reported an inverse relation between melanoma risk and vitamin D intake in the highest quintile (>3.67 mcg/day), and a stronger association in males and in older subjects [92]. Moreover, each 1 μg/day increase in vitamin D intake was associated with a 15% reduced risk (OR = 0.85, 0.74–0.97, *p* = 0.020) [92].

In patients with melanoma, elevated circulating vitamin D levels have been associated with thinner tumors and longer survival [93,94,95] while vitamin D deficiency has been associated with advanced melanoma stage [96,97], poorer survival [94,97,98] and increased tumor thickness [94,95,99]. Similar findings were obtained by a case–control study, analyzing a sample of 137 incident melanoma cases and 99 healthy controls [100], where a statistically significant difference in the median levels of serum vitamin D between melanoma patients and healthy controls (18.0 vs. 27.8 ng/mL, *p* < 0.001) and a higher percentage of vitamin D deficiency (≤20 ng/mL) within cancer patients compared with disease-free ones (66.2% and 15.2%, respectively) were found. Interestingly, vitamin D insufficiency (21–29 ng/mL) versus deficiency (≤20 ng/mL) was found to be significantly inversely associated with melanoma (OR = 0.13, 0.06–0.27, *p* < 0.001), suggesting that the more vitamin D approaches normal values, the lower the OR, and the greater the protective action. Moreover, in a pooled analysis of 25 studies, 25(OH)vit D levels showed an inverse association with Breslow thickness (Standardized Mean Difference (SMD) = 0.24, 0.16–0.33, *p* < 0.001 for ≤1 vs. >1 mm thickness) and mortality rates in melanoma patients (HR = 1.56, 1.26–1.93, *p* < 0.001) [101]. 

In animal models and in vitro studies, vitamin D and its analogues increased the effectiveness of cancer chemotherapy drugs (such as doxorubicin, cisplatinum, gemcitabine, and cyclophosphamide) and sensitized malignant cells to ionizing and proton beam radiation [102]. At present, there is no consensus on the clinical recommendations for vitamin D intake and optimal serum levels in melanoma patients and those at risk for melanoma, although a target serum vitamin D of 70 to 100 nmol/L (28–40 ng/mL) for melanoma patients was proposed [103]. Well-designed RCTs are, therefore, needed to better define the role of vitamin D and its analogues in human melanoma.

#### 4.2.7. Vitamin A

Vitamin A is a group of fat-soluble compounds named retinoids; they include retinol (directly absorbed from animal sources and metabolized from vegetable carotenoids), retinaldehyde, and several pro-vitamins called carotenoids [104]. A meta-analysis of observational studies reported a 20% reduced risk of melanoma in subjects with the highest intake of retinol (OR = 0.80, 95%CI 0.69–0.92) [105], while a null effect on melanoma incidence was reported both for vitamin A (OR = 0.86, 0.59–1.25) and beta-carotene supplementation (OR = 0.87, 0.62–1.20) [105]. In addition, no association between beta-carotene and melanoma risk was reported in a meta-analysis of RCTs (RR = 0.98, 0.65–1.46) [106], but these results were limited due to the availability of only two trials.

#### 4.2.8. Vitamin E

Vitamin E is present in vegetables, oils, seeds, corn, soy, whole wheat flour, margarine, nuts, some meats, and some dairy products [107]. It includes fat-soluble compounds, mainly represented by tocopherols and tocotrienols, with photo-protective and anti-oxidative properties in vitro [107]. 

The Selenium and Vitamin E Cancer Prevention Trial (SELECT) randomized 35,533 men to either selenium (200 ug/d), vitamin E (400 IU/d), both agents, or placebo [108]. The vitamin E group showed a 17% greater risk of prostate cancer after 12-y of follow up (HR = 1.17, 95%CI 1.004–1.36) [108]; no significant effect was reported on other cancer endpoints [109].

A review of intervention and observational studies (including three studies on melanoma) failed to find a protective effect of vitamin E intake and melanoma risk [110]. Indeed, an association was reported only in a nested case–control study on 40,000 Finland subjects, where serum a-tocopherol concentration was negatively associated with melanoma incidence (RR = 0.20, no CI provided, *p* < 0.01) [111]. Accordingly, in a study of 204 melanoma cases and 248 controls, no association between vitamin E intake (including supplements) and melanoma was reported [78]. However, the subjects in the highest quintile of vitamin E intake from food displayed a lower melanoma risk (OR = 0.5, 0.3–0.9, *p* = 0.03), but the statistical model assumed a linear dose–response relationship, which was not well-substantiated [78]. Recently, a genetic-based Mendelian randomization analysis revealed a lack of correlation between the intake of vitamins D, E, and B12, and the risk for five types of cancer, including melanoma [112].

Great attention has been focused on the potential use of natural vitamin E analogues and synthetic derivatives as anticancer drugs and adjuvants. Indeed, in experimental animal models and in vitro, cytotoxic, pro-apoptotic, and anti-proliferative properties on melanoma cells have been demonstrated for alpha-tocopheryl succinate [113], d-δ-Tocotrienol [114,115], the combination of coenzyme Q10 with vitamin E acetate in polymeric nanocapsules [116], and α-tocopheryl linolenate solid lipid nanoparticles [117]. However, it is still unclear whether these compounds might be effective in humans.

#### 4.2.9. Vitamins B 

The vitamin B complex (thiamine, riboflavin, niacin, pantothenic acid, pyridoxine, biotin, folate, and cobalamin) includes water-soluble compounds with different bioactive functions, which are especially present in vegetables and fruits, as well as dairy, meat, legumes, peas, liver, eggs, and fortified grains and cereals [118]. Folate and cobalamin are essential factors for the biosynthesis of nucleotides, DNA replication, and the supply of methyl groups; pyridoxine, thiamine, niacin, riboflavin, biotin, and pantothenic acid are implicated in metabolic and energetic pathways; all contribute to the growth and repair of cells [119]. Folate, pyridoxine, vitamin B_2_, and cobalamin have been related to decreased risk of some cancers, such as breast, lung, and pancreatic cancer [120,121,122]. Indeed, the relationships between B vitamins and the risk of melanoma are highly controversial. Two meta-analyses of RCTs [119,123], both analyzing the same three RCTs on melanoma (*n* = 19,128 adults), reported a significantly reduced risk of melanoma after the supplementation of vitamin B (folic acid, cobalamin, and pyridoxine) (RR = 0.47, 95%CI 0.23–0.94; *p* = 0.03) [119] or folic acid supplementation alone (RR = 0.47, 0.23–0.94) [123]. Conversely, the meta-analysis by Vollset et al. reported 133 cases of melanoma among 3713 involved patients and reported no association between folate supplementation and melanoma risk (RR = 1.04, 0.66–1.64) [124]. 

A more recent cohort study involving 123,834 white US adults, with 1328 documented cases of melanomas over 24–26 years of follow-up, found an increased melanoma risk associated with the highest quintile of folate intake from foods (HR = 1.36, 1.13–1.64, *p* = 0.001), but none for total folate intake (HR = 1.16, 0.96–1.40; *p* = 0.21) [125]. Higher intake of pyridoxine and cobalamin, choline, betaine, and methionine were not associated with the risk of melanoma [125]. Prospective data on 72,308 women from the NHS cohort (1984–2010) and 41,808 men from the HPFS cohort (1986–2010) described a positive association between niacin intake and melanoma risk in men (HR = 1.48, 1.07–2.05), but not in women [126].

### 4.3. Compounds Used as Spices or Supplements

#### 4.3.1. Turmeric

Turmeric (Curcuma longa) is an herbaceous plant of the ginger family (Zingiberaceae), whose natural flavonoid curcumin demonstrated antioxidant, anti-inflammatory, antibacterial, anticancer, insulin-sensitizing, and hypoglycemic properties [127]. In preclinical studies, curcumin was shown to inhibit the growth, invasion, and progression of human melanoma cells by several mechanisms [128,129,130,131] and was demonstrated to overcome chemoresistance in p53-mutant human melanoma cell lines, which are highly resistant to conventional chemotherapeutic agents [132]. The low bioavailability of curcumin limits its utilization as a therapeutic agent; thus, new delivery systems, such as nanoparticles, liposomes, micelles, and phospho-lipid complexes [133,134], as well as curcumin analogues (natural and synthetic molecular analogues or derivatives) were created to improve its anti-cancer effects and bioavailability [135]. However, the clinical use of turmeric should be carefully monitored because it contains other toxic, mutagenic, carcinogenic, and hepatotoxic components, and also exerts inhibitory effects on cytochromes P450, potentially leading to drug interactions [127,134]. 

#### 4.3.2. Ginger

Zingiber officinale Roscoe, commonly referred to as “ginger”, is widely used as a food spice and a medicinal plant in oriental cultures [127]. In vitro, fresh ginger extract showed antitumoral properties by exerting profound cytotoxic effects on a C32 amelanotic melanoma cell line [135]. Another plant of the family, Etlingera elatior, known as “torch ginger”, is rich in several flavonoids that showed anti-cancer activity [136], including a dose- and time-dependent reduction in cell viability in mouse melanoma B16 cells in vitro [136]. 

#### 4.3.3. Ganoderma Lucidum

Ganoderma lucidum, also known as “lingzhi”, is a traditional medicinal fungus, used in in Oriental Countries for its immunomodulatory, anti-inflammatory, antiviral, antioxidative, antiaging, and antitumor properties [137]. In vitro studies reported that G. lucidum polysaccharides may act on melanoma cells both by improving antitumor immunity and with direct antitumor effects, including anti-inflammatory and anti-metastatic activities, via several pathways [137,138,139,140,141,142].

## 5. Conclusions

A large number of studies on melanoma and dietary factors are available, but their results are often controversial, since opposite conclusions are reported. Well-designed, placebo controlled, clinical trials are lacking, while the large amount of available literature refers to observational studies, mostly retrospective, that do not take into consideration the known risk factors for melanoma development (such as sun exposure, skin phenotype/phototype, and number of nevi). Therefore, recommendations for or against specific diets, foods, or supplements for melanoma prevention or modulation cannot be formulated at present. All available studies agreed about the beneficial effects of the MeD and DASH diets, weight loss, and the deleterious role of alcohol (especially white wine). Until more robust evidence becomes available, nutritional counseling to promote healthy dietary behaviors and the maintenance of normal body weight should be the focus of preventive campaigns at the population level. No conclusive evidence is available for several other nutritional factors (such as vegetables, legumes, fruits, cereals, sweets, eggs, processed meat, tea, and vitamins A, B, C, and E) and further well-conducted studies are therefore needed. Experimental animal models and in vitro studies have shown multiple pathways involved in the effector activities of nutritional factors, such as antioxidant action with reduction in oxygen reactive species, anti-inflammatory and immunomodulatory effects with potential immune-protective activities against UV radiation, and cytotoxic, pro-apoptotic, and sensitization effects to chemo- and/or radiotherapy of melanoma cells. Far from being applied in clinical practice, these data could represent the basis for the development of well-designed clinical trials for a complete understanding of the effect of nutrition on the incidence and development of melanoma in order to optimize strategies for both disease prevention and the support of existing treatments.

## Figures and Tables

**Table 1 nutrients-13-04018-t001:** Summary of the main relationships between nutritional factors and melanoma prevention.

Factor	Prevention
Obesity	Increased risk, particularly in males [12,19]
Mediterranean diet	Lower incidence, particularly in <50 years females [35,40]
DASH diet	Lower incidence in women < 50 years [35]
Vegetables and fruit	Lower incidence (small case–control studies), no relation in largest cohort studies [45] Reduced risk for onion, garlic legumes, and increased risk for cabbages in an Italian case–control study [48]
Cereals and sweets	Increased risk in an Italian case–control study [48]
Red and processed meat	Protective effect [49,52]
Eggs	Reduced risk [48]
Olive oil	Reduced risk for intakes up to 60 g/day [48]
Alcohol intake	Increased risk [57,58,59,60], especially due to consumption of white wine [58,59] and liquor [58]
Coffee	Reduced risk [61,62,63,64,65]
Tea	No confirmed association [64,66]. Protective effect in a small case–control study [67]
PUFA	Null or very small effect [76]. Reduced risk with n3-PUFAs [67,73] and increased risk with n-6 PUFAs [74]
Vitamin C	Increased risk with high intakes of vitamin C from food [46], and citrus fruits [79,81]. Protective effect in younger females and phototypes II and III [80] in a small study
Vitamin D	Null [87] or small protective effect in women [90,91] and men [92]
Vitamin A	Reduced risk with high retinol intake [49,105]
Vitamin E	Null effect [110,111,112]
Vitamins B	Controversial associations [119,123,124,125,126]

DASH: Dietary Approaches to Stop Hypertension; PUFAs: polyunsaturated fatty acids.

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
