# Peer review of "Nutritional Interventions for Patients with Melanoma: From Prevention to Therapy—An Update"

_nutrients, 2021, doi:10.3390/nu13114018_

Round 1
Reviewer 1 Report
This is an interesting manuscript covering some relevant points of nutritional schemes, foods, and nutrients as part of the therapeutic role of these approaches in the prevention and progression of melanoma. However, to this reviewer, an important point is missed. The role of dietary advanced glycation end-products has emerged as an important source of activation of the RAGE/AGEs axis. One of the topics covered in this review- Red and processed meat- is a relevant source of dietary AGEs. The role of AGEs and the activation of its receptor RAGE, has emerged as an important contributor in tumor growth and progression including melanoma, and therefore this topic must be covered and discussed at length.
Author Response
Ref.: Manuscript ID: nutrients-1395590 Nutritional interventions for patients with melanoma: from prevention to therapy. An update.
Dear reviewer
Thank you for reviewing our manuscript. We have revised the manuscript according to your comments. Below, we outline our responses to these comments. We have responded to all the suggestions, and we are hopeful that the clarity of the manuscript has been improved, according to the careful and thoughtful comments. The parts of the text which were modified are highlighted in yellow in the present version.
We appreciate your careful evaluation of our work and hope that this revision meets with your approval.
Thank you again for your interest in our work and for giving us the opportunity to improve the quality of our paper.
Best regards,
Pietro Quaglino, on behalf of all authors
Department of Medical Sciences
University of Torino
Answers to the Reviewer #1
Ref.: Manuscript ID: nutrients-1395590 Nutritional interventions for patients with melanoma: from prevention to therapy. An update.
Thank you very much for your kind and useful comments. Please find the answers to your suggestions listed below.
This is an interesting manuscript covering some relevant points of nutritional schemes, foods, and nutrients as part of the therapeutic role of these approaches in the prevention and progression of melanoma.
Thank you very much for these kind comments that are greatly appreciated!
However, to this reviewer, an important point is missed. The role of dietary advanced glycation end-products has emerged as an important source of activation of the RAGE/AGEs axis. One of the topics covered in this review- Red and processed meat- is a relevant source of dietary AGEs. The role of AGEs and the activation of its receptor RAGE, has emerged as an important contributor in tumor growth and progression including melanoma, and therefore this topic must be covered and discussed at length.
According to your request, the role of AGE and the activation of its receptor RAGE in melanoma growth and progression has been described and discussed (page 5, line 196). Thank you for this suggestion.

Reviewer 2 Report
In this review, the authors discuss risk factors for cutaneous melanoma including nevi, skin phenotype/phototype and obesity. They go on to examine a variety of diets, vitamins and spices that have been studied for their effects on melanoma. The conclude that their review provides an update and summary of the effects of nutrition and specific foods on melanoma prevention and progression.
Major points-
- The authors do not define phototype, nor do they the pigmentary phenotypes (red hair color, light eye color, freckles) that increase risk for melanoma. This information is important for clinicians who might be advising patients on how they might lower their risk for melanoma.
- The authors should not refer to the B vitamins as a single entity. They are biochemically complex and their individual roles in biological systems are diverse. The bioactivity of each B vitamin discussed should be summarized separately and explicitly.
- In the section on red meat the authors state that the observed protective effect of diets rich in red meat are consistent with a protective role for nicotinamide against melanoma. This comes from a single sentence in the discussion of reference 48 that refers to a clinical trial that examined of the effects of nicotinamide supplementation on risk for non-melanoma skin cancers (PMID:26488693). That study found no protective effect on risk for melanoma.
- The discussion of the role of vitamin D in melanoma is unacceptable. The photochemical production of Vitamin D3 in the skin is not discussed at all. This makes it impossible for the reader to understand how UV exposure might have some health benefit, and consequently why the effects of vitamin D on melanoma risk are complex.
- Also in the discussion of the role of vitamin D, the authors state that the primary effect of UV on risk for melanoma is the generation of reactive oxygen species (ROS). This is not consistent with copious amounts of experimental data from humans tumors and model systems. The direct absorption of UV photons by DNA result in formation of cyclobutane pyrimidine dimers, which if not repaired properly, result in C>T mutations, which are found in abundance in both melanomas and non-melanoma skin cancers.
- The discussion of vitamin E and its potential role in melanoma prevention does not mention the increased risk of prostate cancer in men found in a randomized placebo controlled Phase III clinical trial.
- Recommendations in the conclusion section are not founded on proper evidence. For example, the authors recommend high coffee intake for melanoma prevention based on observational studies. There is no comment on whether these studies were controlled for risk factors for melanoma. Recommendations for prevention should be based on well-designed placebo controlled clinical trials.
- Also in the conclusion section the authors state that different dietary agents affect melanoma at various stages. They do not define these stages, nor do they describe the experimental evidence for their claims that diet has an influence on the natural history of melanoma by any specific mechanism. Their conclusions are not supported by the data presented.
Minor points-
There are numerous language and typographical errors throughout the manuscript.
Author Response
Ref.: Manuscript ID: nutrients-1395590 Nutritional interventions for patients with melanoma: from prevention to therapy. An update.
Dear Reviewer
Thank you for reviewing our manuscript. We have revised the manuscript according to your comments. Below, we outline our responses to these comments. We have responded to all the suggestions, and we are hopeful that the clarity of the manuscript has been improved, according to the careful and thoughtful comments. The parts of the text which were modified are highlighted in yellow in the present version.
We appreciate your careful evaluation of our work and hope that this revision meets with your approval.
Thank you again for your interest in our work and for giving us the opportunity to improve the quality of our paper.
Best regards,
Pietro Quaglino, on behalf of all authors
Department of Medical Sciences
University of Torino
Answers to the Reviewer #2
Ref.: Manuscript ID: nutrients-1395590 Nutritional interventions for patients with melanoma: from prevention to therapy. An update.
Thank you very much for your kind and useful comments. Please find the answers to your suggestions listed below.
In this review, the authors discuss risk factors for cutaneous melanoma including nevi, skin phenotype/phototype and obesity. They go on to examine a variety of diets, vitamins and spices that have been studied for their effects on melanoma. The conclude that their review provides an update and summary of the effects of nutrition and specific foods on melanoma prevention and progression.
Major points-
1.The authors do not define phototype, nor do they the pigmentary phenotypes (red hair color, light eye color, freckles) that increase risk for melanoma. This information is important for clinicians who might be advising patients on how they might lower their risk for melanoma.
Phototypes and phenotypes have been defined, in accordance with your request (page 1, line 38-42) and ref. 8 has been added. Thank you.
2.The authors should not refer to the B vitamins as a single entity. They are biochemically complex and their individual roles in biological systems are diverse. The bioactivity of each B vitamin discussed should be summarized separately and explicitly.
Thank you for this suggestion. The bio-activity of each vitamin B has been reported shortly, and the specific contribution of each B vitamin has better explained, as follows (page 9, line 404):
“The vitamin B complex (thiamine, riboflavin, niacin, pantothenic acid, pyridoxine, biotin, folate, and cobalamin) includes water-soluble compounds with different bioactive functions especially present in vegetables, fruits, as well as dairy, meat, legumes, peas, liver, eggs, and fortified grains and cereals [118]. Folate and cobalamin are essential factors for the biosynthesis of nucleotides, DNA replication, and supply of methyl-groups; pyridoxine, thiamine, niacin, riboflavin, biotin and pantothenic acid are implicated in metabolic and energetic pathways; all contribute to the growth and repair of cells [119]. Folate, pyridoxine, vitamin B2, and cobalamin have been related to decreased risk of some cancers, such as breast, lung, and pancreatic cancer [120–122]. Indeed, the relationships between vitamins B and the risk of melanoma are highly controversial. Two meta-analyses of RCTs [119,123], both analyzing the same 3 RCTs on melanoma (n=19,128 adults), reported a significantly reduced risk of melanoma after the supplementation of vitamin B (folic acid, cobalamin, and pyridoxine) (RR=0.47, 95%CI 0.23-0.94; p=0.03) [119] or folic acid supplementation alone (RR=0.47, 0.23-0.94) [123]. Conversely, the meta-analysis by Vollset et al reported 133 cases of melanoma among 3713 patients involved and reported no association between folate supplementation and melanoma risk (RR=1.04, 0.66–1.64) [124]”.
3.In the section on red meat the authors state that the observed protective effect of diets rich in red meat are consistent with a protective role for nicotinamide against melanoma. This comes from a single sentence in the discussion of reference 48 that refers to a clinical trial that examined of the effects of nicotinamide supplementation on risk for non-melanoma skin cancers (PMID:26488693). That study found no protective effect on risk for melanoma.
We fully agree with you that previous sentence was misleading. We have now re-written the sentence as follows (page 5, line 187):
“This beneficial effect may be related to the compounds contained in red and processed meat, such as retinol, with an inhibitory effect on promoting melanoma by restoring “normal” functions (differentiation) of melanocytes [49,50], and nicotinamide (NAM), a niacin derivative, with a chemo-preventive effect via an immune response at pharmaco-logical doses in vivo [51].”
4.The discussion of the role of vitamin D in melanoma is unacceptable. The photochemical production of Vitamin D3 in the skin is not discussed at all. This makes it impossible for the reader to understand how UV exposure might have some health benefit, and consequently why the effects of vitamin D on melanoma risk are complex.
Thank you for this comment. We fully agree with you that previous text about vitamin D was unclear. The following sentences have now been added, in accordance with your request (page 7, line 306):
“Vitamin D is a fat-soluble prohormone whose active form, 1,25-hydroxyvitamin D (1,25(OH)2D), is produced after hydroxylation in the liver and kidney [82]. Only about 10-20% of vitamin D requirements derive from diet, since about 80-90% are covered from the photosynthesis in the skin by the action of UV-B radiation [83]. The skin is not only the site of vitamin D synthesis, but also a target tissue for biologically active metabolites of vitamin D [84]. The 1,25(OH)2D binds to the nuclear vitamin D receptors (VDR), which are present in most tissues and cells in the body, including keratinocytes and melanocytes [85]. Besides maintaining serum calcium and phosphorus homeostasis, it plays a role in modulating the immune, cardiovascular, and inflammatory systems, as well as in participating in the regulation of cell proliferation, differentiation, and apoptosis [82]. The bio-logical anti-cancer effects of 1,25(OH)2D include the induction of cell-cycle arrest, stimulation of apoptosis and inhibition of angiogenesis and metastasis [86]. Indeed, the interpretation of vitamin D effects on the risk of skin cancer is complex because the exposure to UV radiation is both the most important source of vitamin D production with its potential anticancer properties and an environmental risk factor implicated in skin carcinogenesis [87] through the formation of cyclobutane pyrimidine dimers bringing DNA replication-dependent mutagenicity [88] and by the production of reactive oxygen species (ROS), leading to premature skin aging and lipid peroxidation [89]”.
5.Also in the discussion of the role of vitamin D, the authors state that the primary effect of UV on risk for melanoma is the generation of reactive oxygen species (ROS). This is not consistent with copious amounts of experimental data from humans tumors and model systems. The direct absorption of UV photons by DNA result in formation of cyclobutane pyrimidine dimers, which if not repaired properly, result in C>T mutations, which are found in abundance in both melanomas and non-melanoma skin cancers.
We agree with the reviewer that the mechanisms leading to DNA damage and UV-mediated tumorigenesis are not only related to the development of ROS, but mainly rely on the formations of pyrimidine dimers. This has been added on page 8 (line 322).
6.The discussion of vitamin E and its potential role in melanoma prevention does not mention the increased risk of prostate cancer in men found in a randomized placebo controlled Phase III clinical trial.
According to your request, we have now mentioned this randomized controlled trial (page 9, line 380).
7.Recommendations in the conclusion section are not founded on proper evidence. For example, the authors recommend high coffee intake for melanoma prevention based on observational studies. There is no comment on whether these studies were controlled for risk factors for melanoma. Recommendations for prevention should be based on well-designed placebo controlled clinical trials.
We fully agree with your suggestion and have now commented about the limitations of the available evidence. The conclusion section has been greatly curtailed (pages 10 and 11, from line 463).
8.Also in the conclusion section the authors state that different dietary agents affect melanoma at various stages. They do not define these stages, nor do they describe the experimental evidence for their claims that diet has an influence on the natural history of melanoma by any specific mechanism. Their conclusions are not supported by the data presented.
The conclusions have been re-written in order to be in line with the available evidence (pages 10 and 11, from line 463). Previous statements which were not supported by experimental evidence have been removed. Thank you.
Minor points
There are numerous language and typographical errors throughout the manuscript.
The text has been carefully checked for language and typographical errors. Many parts have been reformulated or corrected. Thank you.

Round 2
Reviewer 2 Report
The authors have made a significant effort to address the points raised in the review and they have greatly improved the manuscript.